Effects of macroalgae loss in an Antarctic marine food web: applying extinction thresholds to food web studies

Cordone Georgina gcordone@cenpat-conicet.gob.ar 1
Marina Tomás I. 2 3 4
Salinas Vanesa 2
Doyle Santiago R. 2 3
Saravia Leonardo A. 2 3
Momo Fernando R. 2 3
1 Centro Nacional Patagónico (CCT CONICET-CENPAT), Centro Para el Estudio de Sistemas Marinos (CESIMAR) , Puerto Madryn , Chubut , Argentina
2 Universidad Nacional de General Sarmiento, Instituto de Ciencias (ICI) , Los Polvorines , Buenos Aires , Argentina
3 Universidad Nacional de Luján, Instituto de Ecología y Desarrollo Sustentable (INEDES) , Luján , Buenos Aires , Argentina
4 Consejo Nacional de Investigaciones Científicas y Técnicas (CONICET), Centro Austral de Investigaciones Científicas (CADIC) , Ushuaia , Tierra del Fuego , Argentina
Sarmento Hugo
Electronic publication date: 2018 Sep 12
Publication date: 2018
Volume: 6
Electronic Location ID: e5531
Received 2018 Mar 2; Accepted 2018 Aug 7
Copyright: ©2018 Cordone et al.
Copyright year: 2018
Copyright holder: Cordone et al.
License: This is an open access article distributed under the terms of the Creative Commons Attribution License, which permits unrestricted use, distribution, reproduction and adaptation in any medium and for any purpose provided that it is properly attributed. For attribution, the original author(s), title, publication source (PeerJ) and either DOI or URL of the article must be cited.
License URL: https://creativecommons.org/licenses/by/4.0/

Keywords: Antarctic, Food webs, Extinctions, Macroalgae loss

Funding: Consejo Nacional de Investigaciones Científicas y Técnicas (CONICET, Argentina) PIO 14420140100035CO Universidad Nacional de General Sarmiento and Alfred Wegener Institute for Polar and Marine Research (AWI, Germany) Marie Curie Action IRSES FP7 IRSES, Action No. 319718 This research was supported by Consejo Nacional de Investigaciones Científicas y Técnicas (CONICET, Argentina), Universidad Nacional de General Sarmiento and Alfred Wegener Institute for Polar and Marine Research (AWI, Germany). The work was partially funded by PIO 14420140100035CO CONICET Argentina and conducted in the frame of GC’s Ph.D. studies, whose scholarship (CONICET, Argentina) supported the rest of the study. The work was conducted in the frames of the EU research network IMCONet funded by the Marie Curie Action IRSES (FP7 IRSES, Action No. 319718). There was no additional external funding received for this study. The funders had no role in study design, data collection and analysis, decision to publish, or preparation of the manuscript.

==============================
Antarctica is seriously affected by climate change, particularly at the Western Antarctic Peninsula (WAP) where a rapid regional warming is observed. Potter Cove is a WAP fjord at Shetland Islands that constitutes a biodiversity hotspot where over the last years, Potter Cove annual air temperatures averages increased by 0.66 °C, coastal glaciers declined, and suspended particulate matter increased due to ice melting. Macroalgae are the main energy source for all consumers and detritivores of Potter Cove. Some effects of climate change favor pioneer macroalgae species that exploit new ice-free areas and can also decline rates of photosynthesis and intensify competition between species due to the increase of suspended particulate matter. In this study, we evaluated possible consequences of climate change at Potter Cove food web by simulating the extinction of macroalgae and detritus using a topological approach with thresholds of extinction. Thresholds represent the minimum number of incoming links necessary for species’ survival. When we simulated the extinctions of macroalgae species at random, a threshold of extinction beyond 50% was necessary to obtain a significant number of secondary extinctions, while with a 75% threshold a real collapse of the food web occurred. Our results indicate that Potter Cove food web is relative robust to macroalgae extinction. This is dramatically different from what has been found in other food webs, where the reduction of 10% in prey intake caused a disproportionate increase of secondary extinctions. Robustness of the Potter Cove food web was mediated by omnivory and redundancy, which had an important relevance in this food web. When we eliminated larger-biomass species more secondary extinctions occurred, a similar response was observed when more connected species were deleted, yet there was no correlation between species of larger-biomass and high-degree. This similarity could be explained because both criteria involved key species that produced an emerging effect on the food web. In this way, large-biomass and high-degree species could be acting as source for species with few trophic interactions or low redundancy. Based on this work, we expect the Potter Cove food web to be robust to changes in macroalgae species caused by climate change until a high threshold of stress is reached, and then negative effects are expected to spread through the entire food web leading to its collapse.

Introduction

The Western Antarctic Peninsula (WAP) is the region of Antarctica most affected by climate change, with a regional warming rate that doubles what is observed in other Antarctic regions (McClintock, Ducklow & Fraser, 2008; Turner et al., 2009; Bromwich et al., 2013; Nicolas & Bromwich, 2014). One particular example of environmental changes in WAP was observed at Potter Cove (Fig. 1), a hotspot of biodiversity, where over the last years the average winter air temperatures increased by 0.66 °C, coastal glaciers declined and suspended particulate matter increased due to ice melting (Schloss et al., 2012; Bers et al., 2013; Ducklow et al., 2013; Quartino et al., 2013; Grange & Smith, 2013; Lagger et al., 2017). These changes produce a variety of effects on the organisms of the cove; pioneer macroalgae species are favored by the increase of rocky bottom areas due to glaciers retreat (Quartino et al., 2013), whereas increased suspended particulate matter affects rates of photosynthesis and intensity of competition between species (Quartino et al., 2013; Deregibus et al., 2016). Moreover, a long-term succession study showed that sites with higher levels of stress and disturbance caused by glacial influence presented lower number of macroalgal taxa and a tendency to decrease diversity over time (Campana et al., 2018). In recent years, environmental changes have triggered modifications at the benthic community of Potter Cove (Pasotti et al., 2015; Sahade et al., 2015). The community structure changed from an assemblage dominated by filter feeders squirts to a mixed assemblage, and this alteration can be explained by organism sensitivity to increased sedimentation (Sahade et al., 2015). A descriptive analysis of the Potter Cove food web has suggested fragility to species loss and susceptibility to propagation of trophic cascade effects based on topology parameters as connectance (C) or link density (LD) (Marina et al., 2018). However, alternative food sources could contribute to food web robustness (Marina et al., 2018).

Figure 1 Map of Potter Cove at 25 de Mayo/King George Island.

Map of Potter Cove at 25 de Mayo/King George Island where scientific research station Base Carlini is highlighted (Map data: ©2018 Google Earth, DigitalGlobe). 25 de Mayo/King George Island is the largest island of the South Shetland Islands, lying 120 km from the Western Antarctic Peninsula (WAP).

About forty species of macroalgae have been reported at Potter Cove (Quartino & De Zaixso, 2008a), a significant fraction of the approximately 130 registered for Antarctica as a whole. In Potter Cove, macroalgae constitute a direct pathway of energy and matter into organisms that feed on them, and also indirectly, through species that feed on macroalgae detritus (Quartino, Boraso de Zaixso & Momo, 2008b). Macroalgae are not only important as primary producers that supply significant amounts of fixed carbon to marine ecosystems, but as substrates for organisms that inevitably provide a large fraction of the secondary production to the benthos (Gómez et al., 2009). Considering this, we can reaffirm the words of Neushul (1965) that macroalgae are the energetic base of Potter Cove food webs.

Since the macroalgal community plays a fundamental role in the Potter Cove food web, it might affect ecosystem dynamics and stability through propagation of direct and indirect effects. Changes in species abundance or presence can indeed alter the whole community structure (e.g., Dulvy et al., 2000; Jackson et al., 2001; Ellison et al., 2005; Sorte et al., 2017). Moreover, the loss of species can lead to a cascade of secondary extinctions that propagate through the community (Ebenman, Law & Borrvall, 2004; Sanders, Sutter & Veen, 2013; Donohue et al., 2017; Witman, Smith & Novak, 2017). Food webs are considered an excellent tool to explore possible effects of global change and biodiversity loss on communities and ecosystems, since trophic interactions are crucial for the survival of both resource and consumer species (Ings et al., 2009; Bascompte, 2009; Hastings, McCann & De Ruiter, 2016). The vulnerability of food webs to adverse effects can be estimated from the topology of the network (De Santana et al., 2013). Several authors have studied the effects of species extinctions in food webs (Sole & Montoya, 2001; Dunne, Williams & Martinez, 2002; Memmott, Waser & Price, 2004; Allesina, Bodini & Bondavalli, 2006; Curtsdotter et al., 2011; Eklöf, Tang & Allesina, 2013; Bellingeri & Bodini, 2013). Borrvall et al. (2000) studied the effect of omnivory and redundancy in model food webs, and found that both lessen the risk of cascading extinctions following primary loss of species. Additionally, Montoya & Solé (2002) demonstrated that food webs are vulnerable to the elimination of the most connected species. Moreover, it has been observed that extinctions and invasions can alter costal marine food webs by reshaping trophic level relationship (Byrnes, Reynolds & Stachowicz, 2007). In this sense, it is not trivial to study the behavior of food webs to local species extinctions since it can be a suitable proxy of community response to alterations.

Although much is known about macroalgae species and the environmental changes that Potter Cove is facing, there is a lack of information of how local extinctions of macroalgae species might impact on other species of the Cove and on the community as a whole. In this work we study the robustness of Potter Cove food web by simulating primary loss of macroalgae species using a topological approach. We hypothesized that the loss of these species from Potter Cove food web will generate a cascade of secondary extinctions that will lead the food web to collapse by a bottom-up mechanism. The aim of this study was to explore the robustness of Potter Cove food web to local extinctions of macroalgae species as possible consequences from ongoing climate change.

Material and Methods

Study site

Potter cove is a WAP fjord of 4 km long and 2.5 km wide, on the southern coast of 25 de Mayo/King George Island (62°14′S, 58°40′W), the largest of the South Shetlands Archipelago (Fig. 1). It is adjacent to Maxwell Bay, which connects to the Bransfield Strait. Water circulation in Potter Cove is strongly influenced by the general circulation of Maxwell Bay (Roese & Drabble, 1998). The fjord is divided into an outer and an inner Cove which differ in bottom characteristics (Klöser et al., 1994; Klöser, Quartino & Wiencke, 1996). The outer Cove consists of hard substrate of solid rocks and big boulders, whereas the inner part is dominated by soft sediments with high presence of muddy substrate. The rocky shores of the outer Cove are colonized by a large biomass of macroalgae (Klöser, Quartino & Wiencke, 1996; Quartino, Zaixso & Boraso de Zaixso, 2005), while the inner Cove has one of the highest concentrations of benthic filter feeders found in Antarctic coastal areas representing a hotspot of biodiversity (Sahade et al., 1998; Tatián et al., 1998; Tatian, Sahade & Esnal, 2004; Grange & Smith, 2013).

Food web dataset

We based our analysis on the food web presented by Marina et al. (2018). Potter Cove food web includes 91 trophic species comprising algae, amphipods, isopods, sponge, gastropods, bivalves, echinoderms, fishes, among others, and compromising 308 trophic interactions, with a mean trophic level of 2.1 (±0.9) and a maximum trophic level of 4.27 (Fig. 2). It is important to note that, within the 91 trophic species, 24 correspond to macroalgae species and two to detritus sources, fresh and aged detritus. This food web model was constructed based primarily upon studies within the framework of international research cooperation between Argentina and Germany initiated in 1994 (Wiencke et al., 1998; Wiencke et al., 2008). Further details on the construction and properties of the network can be found in Marina et al. (2018).

Figure 2 Common-enemy graph and Potter Cove food web.

(A) Common-enemy graph of Potter Cove food web, nodes represent basal species and links indirect interactions (shared predators). Node and link widths are proportional to number of shared predators. (B) Potter Cove food web, vertical position indicates trophic level and node widths are proportional to total degree (in and out) (Marina et al., 2018). Node colors represent functional groups.

Common-enemy graph

The analysis of secondary graphs is a useful tool for the study of food webs. Any food web can be associated with two non-directed graphs: the niche overlap graph and the common-enemy or resource graph. The former is constructed by linking consumers that share at least one prey in common, and the latter is constructed by linking prey that share at least one consumer in common (Bersier, 2007). Common-enemy graphs provide a representation of indirect interactions within species of the food web. One of these possible indirect interactions is apparent competition, where an increase in prey abundance produces a decrease in another prey of a shared predator, not by direct competition but through an increase in the abundance of this shared predator (Holt & Lawton, 1994). Additionally, the common-enemy graph is useful to know the number of predators that each species share and to compare this number with the degree obtained from the original food web. If a positive correlation exists; we expect a low number of secondary extinctions because more connected species also share a large number of predators. Also, common-enemy graphs represent an approximation to the complexity of the food web because they describe the map of indirect interactions that remain between preys. Such interactions may trigger cascading effects that involve positive and negative impacts on the abundance of prey populations (Holt & Lawton, 1994). The common-enemy graph for Potter Cove food web was obtained by multiplying the original predation matrix (or adjacency matrix as it is classically known in food web literature) by its transpose, considering only those nodes that represent basal species (i.e., species without preys).

Extinctions by topological approach

The method used in this study is based on the topological approach of secondary extinctions (Albert, Jeong & Barabasi, 2000) with the incorporation of extinctions thresholds (Schleuning et al., 2016; Bellingeri & Bodini, 2013). The topological approach focuses exclusively on the presence or absence of consumer-resource relationships, taking into account only the qualitative network structure (i.e., its topology). In this manner, secondary extinctions occur when one consumer species lose all of its prey species (Sole & Montoya, 2001). This approach has the advantage of requiring only the network structure as input, simplifying its application to large complex networks. However, it has its limitations due to the fact that species are lost secondarily only if all of its preys are removed. It is also assumed that all species have the same baseline probability of extinctions, whereas in natural systems some species are more vulnerable than others (Eklöf, Tang & Allesina, 2013). One possibility to reduce such limitations is incorporate thresholds of extinctions that allow to model secondary extinctions of species which have not necessarily lost all of their preys. Schleuning et al. (2016) applied this approach to mutualistic networks, while Bellingeri & Bodini (2013) did it in food webs. Bellingeri & Bodini (2013) define threshold (v) as the minimum level of energy necessary for species’ survival. After each node removal, the fraction of original incoming energy e(i) is calculated for each species, and the species i is secondarily lost if this fraction is equal or less than the threshold (i.e., e(i) ≤ v). For example, a value of v = 0.25 means that species i goes extinct if e(i) is equal or less than 25%, when the 75% of the inflow is lost. In the classical topological approach, v is implicitly assumed to be equal to 0, and a species goes extinct when its energy inflow is null (Bellingeri & Bodini, 2013). Since the Potter Cove food web lacks energy flow information, it is necessary to assume that all preys contribute equally to its predator diets and redefine v concept as the fraction of incoming links l(i) from original ones that species i needs to survive after a primary loss (i.e., species i is considered secondary loss when l(i) ≤ v). We performed diversity loss simulations of macroalgae species (and detritus) of the Potter Cove food web. Simulations consisted in removal of target species from the predation matrix by deleting row and columns (primary extinction). Following each primary extinction, we looked for secondary extinctions in a recursive manner as follows. We computed an intermediate predation matrix where each l(i) was calculated and compared to the set extinction threshold. If l(i) was equal or less that v, species i was removed from the intermediate predation matrix, thus leading to a secondary extinction, and a new intermediate matrix was computed. This procedure was repeated until no l(i) was less than v and a final predation matrix was obtained. Then the final predation matrix was characterized by computing the number of species (S), the number of trophic interactions or links (L) and connectance (C = L∕S2), (Pimm, 2002; Cohen, 1989), and was used as an input for a new primary loss simulation. This procedure was repeated until all macroalgae species were removed (an explicit example is presented in: Fig. S1). Since the extinction procedure is carried out until all secondary extinctions are recorded, all the remaining species in the food web are connected to at least one basal species (i.e., they had at least one path that connected them to a basal producer). It is important to note that there are other basal species in Potter Cove food web different from macroalgae species. Therefore a non-null final predation matrix is possible given that there are species that depend on other sources (e.g., phytoplankton).

Macroalgae species were removed from the predation matrix according to different removal sequences established by the degree of each macroalgae (total number of trophic interactions per node) and total biomass in Potter Cove. Four sequences of extinctions were used: random order, degree in ascending and descending order, and biomass order in ascending order. The sequences established by the species’ degree depended only on the food web structure, and in consequence all macroalgae species could be eliminated following these sequences. According to Sole & Montoya (2001), we expected that the descending order sequence presented the more abrupt changes, and eliminations in random order set an intermediate situation between ascending and descending order. Thus, the upper and lower limits in which the parameters of the network could vary are established by descending and ascending order. We decided to eliminate macroalgae from lower values of biomass to higher values (i.e., in ascending order of biomass) since extinction risk is typically high for rare species because small populations are more vulnerable to environmental and demographic stochasticity than larger ones (Lawton, 1994; McKinney, 1997). In this way, we eliminated 20 macroalgae species based on biomass data obtained by Quartino & De Zaixso (2008a). Extinction simulations were performed varying v (0 %, 25%, 50% and 75%), and 100 repetitions were randomly performed for the sequences: random, ascending and descending degree orders. As a consequence, we had four different responses per topological parameter (L vs primary extinctions, C vs primary extinctions and secondary extinctions vs primary extinctions). After the loss of all macroalgae species, we simulated the removal of the two detritus sources (fresh and aged) given that macroalgae species are the main source of these types of detritus in Potter Cove ecosystem (Quartino, Boraso de Zaixso & Momo, 2008b). We tested whether there was a correlation between large-biomass and high-degree macroalgae species by computing the Pearson’s correlation coefficient. The impact of secondary extinctions in the food web, was assessed considering the percentage of species loss by secondary extinction with respect to maximum number of possible extinctions (i.e all non-basal species) which corresponded to 61 species. The v complement (100% - v) indicates the percentage of prey that a predator has to lose to go extinct. We compared the v complement with the percentage of secondary extinctions in order to assess the effect of macroalgae loss in the whole food web. A percentage of secondary extinction greater than the v complement indicates a collapse of the food web produced by the instability of the network and not because of individual effects.

Testing thresholds

To test the importance of v in secondary extinctions, we simulated the primary loss of a given quantity of macroalgae species choosing at random the identity of them. We varied v from 5 to 95% by 5%, and recorded the number of secondary extinctions for each v value. These simulations were performed increasing the number of macroalgae primary losses: three, five, 10, 15 and 20 species. We obtained mean and confidence interval values for the five quantities analyzed. Then, we plotted secondary extinctions vs v.

The GNU R software (R Core Team, 2017, version 3.4.2) was used for food web simulations and matrix computations, and Visone (Brandes & Wagner, 2004, version 2.9.2) was used to plot Potter Cove food web and the common-enemy graph.

Results

Common-enemy graph

The common-enemy graph of basal species from Potter Cove food web had all species connected with a high density of 9.53 interactions per node (Fig. 2). It compromised 30 species consisting of: 24 macroalgae species, two sources of diatoms (i.e., benthic and epiphytic), one node representing phytoplankton and 3 non-living nodes corresponding to fresh, aged detritus and necromass. Among all the links of the graph (286 indirect interactions), most of them (222) were between macroalgae species. However, fresh detritus and benthic diatoms shared the maximum number of common predators (11). There was a high correlation between the degree of each macroalgae species obtained from the original food web and the number of shared predators calculated from the common-enemy graph (Pearson’s correlation coefficient 0.87).

Extinctions by topological approach

As the number of primary extinctions increased, L of the remaining food web decreased irrespectively of v (Fig. 3). Removal in a descending order, from highest to lowest degree, showed a fast loss of L at the beginning; while the removal in ascending order showed a slow loss of L. Random order resulted in an intermediate position between ascending and descending degree sequences. Elimination of macroalgae species based on biomass order displayed a response very similar to the response of the removal in ascending degree order at first, and then it behaved as the random extinctions sequence with a similarity that increased with v. There was a low correlation between macroalgae biomass and degree (Pearson’s correlation coefficient 0.27), which means macroalgae of high biomass did not necessarily have a high degree in Potter Cove food web. When we compared the responses at different v, we found that an increase in v led to an increase in the number of lost interactions (Lmin values of 211, 197, 129, and 30 for v of 0, 25, 50, and 75%, respectively).

Figure 3 Links response to simulation of species loss.

Remaining links (L) vs primary extinctions, when macroalgae species and detritus were removed. L was recorded after primary and secondary extinctions have taken place. Each box corresponds to a different threshold (v) ((A) 0%, (B) 25%, (C) 50% and (D) 75%). Macroalgae loss was done following different sequences of extinctions: red, ascending degree (mean and standard deviation); blue, descending degree (mean and standard deviation); violet, random degree (mean and standard deviation) and green, biomass (ascending order).

The response of C to extinctions differed from L response pattern (Fig. 4). At low v (0% and 25%), C increased with primary extinctions. Regarding this, removal in descending order led to a slow increment of C. On the other hand, the loss of low connected species was associated with a high slope in the curve, meaning that the elimination of species by ascending degree led to a rapid increase of C. Random removal depicted an intermediate situation between ascending and descending degree sequences. Elimination based on biomass order displayed a similar response to that of removal in ascending order. An increase in C is directly associated to the way it is computed, as possible interactions (S2) increased faster than real interactions (L). At high v (50% and 75%), C increased for all sequences but showed fluctuations.

Figure 4 Connectance response to simulation of species loss.

Connectance (C) vs primary extinctions, when macroalgae species and detritus were removed. C was computed after primary and secondary extinctions have taken place. Each box corresponds to a different threshold (v) ((A) 0%, (B) 25%, (C) 50% and (D) 75%). Macroalgae loss was done following different sequences of extinctions: red, ascending degree (mean and standard deviation); blue, descending degree (mean and standard deviation); violet, random degree (mean and standard deviation) and green, biomass (ascending order).

We observed a general trend of increase of secondary extinctions with the number of primary losses (Fig. 5), and also with v (6.5, 9.8, 29.5, and 70.5% of secondary extinctions with v of 0, 25, 50, and 75%, respectively). This increase of secondary extinctions is expected because species lose a greater number of preys as the number of incoming links decrease, which leads l(i) to be less than v. Among the elimination sequences, we observed that elimination in descending degree reached maximum number of secondary extinctions in less number of steps than the others sequences and the elimination in ascending degree was the last sequence to reach it. Biomass and random sequences exhibited an intermediate response between the ascending and descending degree order, overlapping in several points.

Figure 5 Secondary extinctions response to simulation of species loss.

Secondary extinctions vs primary extinctions, when macroalgae species and detritus were removed. Each box corresponds to a different threshold (v) ((A) 0%, (B) 25%, (C) 50% and (D) 75%). Macroalgae loss was done following different sequences of extinctions: red, ascending degree (mean and standard deviation); blue, descending degree (mean and standard deviation); violet, random degree (mean and standard deviation) and green, biomass (ascending order).

Testing thresholds

While increasing v value, the number of secondary extinctions increased independently of the number of primary extinctions. In all cases, there was an inflection point in v value that triggered a great number of secondary extinctions. Such inflection point appeared at a lower v value as the number of primary loss increased. It is worth noting that, for all cases, this trigger was generated at a v value greater than 50% (Fig. 6).

Figure 6 Effect of threshold in secondary extinctions.

Secondary extinctions with threshold (v) variations (from 5% to 95% by 5%) when macroalgae are primary lost (blue: three, brown: five, green: ten, red: fifteen, pink: twenty). Points are means of secondary extinctions when x number of primary extinctions took place at a particular v (by random simulations). Shaded area represents confidence interval of the series.

Discussion

We found that the Potter Cove food web is relatively stable to macroalgae loss. A significant number of secondary extinctions were obtained beyond a 50% threshold (v). Indeed, to observe a real collapse where the percentage of secondary extinctions is greater than v complement, a v of 75% was needed. A v of 75% means predators cannot survive if they have less than 75% of their original prey set, a very extreme and unreal condition. Bellingeri & Bodini (2013) showed that reducing only 10% of prey intake, a disproportionate growth of secondary extinctions emerged. Allesina, Bodini & Bondavalli (2006) also found a similar result imposing link thresholds equal to 15%. Our results do not support the hypothesis that the elimination of macroalgae species from the Potter Cove food web will generate a cascade of secondary extinctions, and thus suggests that the Potter Cove food web could be more robust than similar ecological networks (Allesina, Bodini & Bondavalli, 2006; Bellingeri & Bodini, 2013). We found that macroalgae of high degree were positively correlated with those that shared more predators, which indicates that these macroalgae species are functionally redundant (i.e., species with equivalent trophic interactions). Functional redundancy has important consequences in potential cascade extinctions, since it increases food web resistance by means of availability of alternative preys (Borrvall et al., 2000). Functional redundancy and omnivory explain why a high v has to be reached to observe a high number of secondary extinctions. Functional redundancy and omnivory could be the result of a long evolution time and extreme conditions of the studied ecosystem where the preferred strategy is to be a generalist (i.e., species with many feeding links). Redundancy appears to be the rule in many food webs, with stabilizing effects (Lawton & Brown, 1994; Borrvall et al., 2000; Worm & Duffy, 2003; Thompson et al., 2007; Sanders et al., 2018).

The common-enemy graph has a high density value of interactions per node. Similar values were obtained in studies of rocky intertidal ecosystems in Patagonia (Fueyo Sánchez, 2013). The presence of subnets in common-enemy graphs is usually associated with the identification of habitat structures (Holt & Lawton, 1994). The common-enemy graph for the Potter Cove food web exhibited no subnets, consistent with the marine nature of this food web (Marina et al., 2018). Given the high density of the common predators network, it is very difficult to predict indirect relationships between species, as each interaction involves positive and negative effects between species abundances (Holt & Lawton, 1994). The high density of this graph also suggests the presence of high redundancy in the food web.

Montoya & Solé (2002) have shown that the elimination of the most connected species causes the greatest number of secondary extinctions. As expected, removing the most connected macroalgae from the Potter Cove food web produced a fast secondary species loss, and the same happened for the number of links (L). Connectance (C) increased with extinction for all deletion sequences, which is explained by the number of possible interactions (S2) decreasing faster than the number of real interactions (L). However, the number of secondary extinctions remained low even when v was quite high (75%). Analyses of potential secondary extinctions showed that marine food webs appear to be robust to the loss of the most connected species as well as random species (Dunne, Williams & Martinez, 2004). It is important to note that the elimination of species in biomass order generated a similar general response in L, C and secondary extinctions than the loss of the most connected macroalgae species, especially at the first removals and when v was increased. Nevertheless, there was no correlation between large-biomass and high-degree species. This similarity response could be explained because both criteria involved key species that produced an emerging effect on the food web. This means that large-biomass and high-degree species are acting as source for species with few trophic interactions or low redundancy, and this in turn produces a great number of secondary extinctions.

Even though the number of secondary extinctions was low, a clear impact on the emergent properties of the network such as L and C was observed when macroalgae species were lost. These changes could have a significant impact on the fragility and resilience of Potter Cove food web altering the dynamics of the system. Dynamical analyses of the response of communities to species loss usually predict a higher number of secondary extinctions than topological ones (Eklöf & Ebenman, 2006). The main difference between topological and dynamical approaches is that the topological analyses cannot detect indirect effects, as top-down extinctions caused by the loss of a top predator, while dynamical analyses can detect this type of indirect effects (Ebenman & Jonsson, 2005). In this way, the topological approach underestimates the actual number of secondary extinctions, and in consequence overestimates food web robustness. However, the topological approach enables the analysis of more complex food webs, since it only requires knowledge of network structure (Eklöf, Tang & Allesina, 2013). Dynamical analyses have also found that the effect of interaction strengths has important consequences on food web stability (Allesina, Bodini & Bondavalli, 2006; Bellingeri & Bodini, 2013). Furthermore, we have to consider that this food web includes all recorded trophic interactions, though trophic interactions do not always occur simultaneously in time and space. We cannot ignore the additional potentially confounding effects of seasonality and spatial sampling (Ings et al., 2009). In fact, the network represents an ideal niche space and dynamic stability depends more on how interactions materialized in time and space. Some nodes are clearly pulsatile, such as the massive influx of Krill (Aguirre, 2015), which it is difficult to reflect in a static description of the network topology. The consideration of all these factors might increase the fragility of the Potter Cove food web. By reducing the number of possible interactions, predators cannot switch to alternative preys, depending solely on preys that are present and available. For these reasons, future studies considering the effect of time and space, massive pulses of certain species and the strength of each interaction are required to satisfactory predict the impacts of climate change on the currently threatened Antarctic marine ecosystem of Potter Cove.

Conclusions

We found that the Potter Cove food web was relatively robust to local extinctions of macroalgae species under a topological approach considering extinction thresholds (v). We could attribute this robustness to the effects of omnivory and functional redundancy (Borrvall et al., 2000). This robustness becomes clear in the high link density of the common-enemy graph. Therefore, we expect the Potter Cove food web to be moderately affected by the changes in macroalgae species caused by climate change until a high threshold of stress is reached, and then negative effects are expected to spread through the entire food web leading it to collapse. In this manner, our results suggest that, despite the apparent robustness of the Potter Cove food web, structural changes in the community might not be detected until most of the macroalgae species are affected.

Supplemental Information

Figure S1 Scheme of secondary extinctions detection

Scheme of the methodology used to detect secondary extinctions after the removal of a particular species (primary loss). In this example, the extinction threshold (v) was 0.75 and H2, P1 and P2 species were secondarily extinct when their fraction of incoming links l(i) was lower than v after the removal of species A3. A1, A2 and A3 represent algae species, H1 and H2 herbivores, and P1 and P2 predators.

Click here for additional data file.

Supplemental Information 1 Potter Cove food web

Raw data of Potter Cove food web. The adjacency matrix contains the same trophic species in rows (i) and columns (j) where the elements may be only one or zero ( aij = 0 or aij = 1). A one represents the existence of a trophic interaction where the j-species feeds on the i-species.

Click here for additional data file.

Supplemental Information 2 Code to perform macroalgae loss in Potter Cove food web

Code implemented in R to carry out primary extinctions of macroalgae species. A vector with thresholds (v) was compared with the number of prey of each species to detect secondary extinctions.

Click here for additional data file.

We thank Kevin Sánchez and Florencia Ríos for suggestions on language aspects, which helped us to improve the manuscript. We also thank the editor and the two reviewers for their valuable comments which helped us improving the quality and clearness of this paper.

Additional Information and Declarations

Competing Interests

Author Contributions

Data Availability

The authors declare there are no competing interests.

Georgina Cordone, Tomás I. Marina and Vanesa Salinas conceived and designed the experiments, performed the experiments, analyzed the data, contributed reagents/materials/analysis tools, prepared figures and/or tables, authored or reviewed drafts of the paper, approved the final draft.

Santiago R. Doyle and Leonardo A. Saravia conceived and designed the experiments, contributed reagents/materials/analysis tools, authored or reviewed drafts of the paper, approved the final draft.

Fernando R. Momo conceived and designed the experiments, contributed reagents/materials/analysis tools, prepared figures and/or tables, authored or reviewed drafts of the paper, approved the final draft.

The following information was supplied regarding data availability:

The code to perform the extinctions is provided as a Supplemental File.

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
