# Peer review of "Effects of macroalgae loss in an Antarctic marine food web: applying extinction thresholds to food web studies"

_PeerJ, doi:10.7717/peerj.5531_

## Round 0.1 · original submission · Major Revisions

I have received the comments of two reviewers on your paper.

Although all of them highlight the relevance of the study and consider that it involves valuable information, they find several aspects that should be improved in the manuscript. Based on the comments of both reviewers, the decision is to reconsider the publication of the paper after major revisions.

In this way, I invite you to send a revised version of the manuscript. I also recoment that you send the manuscript to English revision before submition. Please make sure that you reply point by point to each comment.

Looking forward to receiving your revised manuscript,

Best regards,

Reviewer 1 ·

Basic reporting

The major claim of this paper is to investigate how the Potter Cove food web respond to loss of macroalgae species, as a consequence from ongoing climate change, a question of high importance and relevance. The authors use a well-defined method, so called topological approach, and conclude that the community is highly robust to extinctions of the major source input, i.e. macroalgae. The manuscript is well structured but less well written. I have a number of comments (major and minor) that I believe should be addressed before the manuscript is considered for publication.

Comments for improvements:
1) The English language needs to be improved throughout the entire manuscript. Examples of such improvements are presented below(strikethrough shows words suggested to be removed, and italic underlined words are suggested to be added):
a. Row 20. Over the last years, Potter Cove annual air temperatures averages increased with 0.66°C.
b. Row 31-33. This result is dramatically different from that found in other food webs where by simply reducing 10% of prey intake caused a disproportionate increase growth of secondary extinctions was observed.
c. Row 43. Antarctica is one of the regions most seriously…
d. Row 86-88. Borvall et al. (2000) studied the effect of omnivory and redundancy in model food webs and found that both lessen the risk of cascading extinctions following further primary loss of species.
e. Row 96-98. For this these reason, we proposed to study the robustness of Potter Cove food web by simulating primary loss extinctions of macroalgae species using a topological approach.
f. Row 147. In this manner, secondary extinctions occur are produced when one consumer species lose losses all of its preys prey species.
g. Row 322. “…massive pulses of certain species and the strength of each interaction are required to satisfactory predict the impacts of climate change on the currently threatened Antarctic marine ecosystems of Potter Cove”.

2) I cannot find the corresponding figure texts.

3) Please link the result sections to the described analysis in the method part using the same subheadings.

4) Check all the references, some of them lack information of Journal volume, issue, and page number. Also go through points and comma.

5) Perhaps the terminology of adjacency matrix could be renamed with the term predation matrix, which is more commonly used in the field of community ecology.

6) Consider changing “detritus” in row 25 to detritivores (as I believe detritus do not itself need energy).

7) Explain the meaning of thresholds of extinction in the summary part.

8) In the summary and on row 302 the authors write that the similarities in the results between the two extinction sequences “ascending biomass” and “descending degree” could be explained by the involvement of key species which have an emerging effect on the food web. Could the authors discuss more about this? Any ideas of common characteristic of these “key species” that could trigger the same food web response when lost?

9) Please add more information to the figure axis. For example the x-axis in fig 3-5 could be named Primary extinction (%).

10) Please add more information of the food web under row 116 Food web dataset. How many species does the food web of Potter Cove consist of? Range of trophic levels in the food web?

11) Consider change the meaning on row 221 “It was observed that when we increased the number of secondary extinctions,…” to “It was observed that when the number of secondary extinctions increased,…”

12) Incomplete meaning on row 244 “The analysis of secondary extinctions showed a general increase trend with threshold (0% threshold, secondary extinctions=6.5 %; 25% threshold, secondary extinctions=9.8%; 50% threshold, secondary extinctions=29.5%; 75% threshold, secondary extinctions=50.6%), and a logical increase with primary extinctions (Fig. 5).”
a. Add to the first part of this sentence “The analysis of secondary extinctions showed a general increase trend… -> in what?.
b. Even though it is a logical response of increased number of secondary extinctions following primary losses, explain this logical part.

Experimental design

Comments for improvements:
1) The authors needs to carefully describe their definition of thresholds used in the study. In the method part (row 158), they write that a “threshold represents the amount of energy and matter intake that triggers the extinction of one species.
a. Here, please describe how energy intake trigger secondary extinctions in a topological model where secondary extinctions are caused by loss of prey-species (i.e. loss of energy intake).
b. Furthermore, the authors write on row 159-161 that “A threshold of 0% corresponds to the classical topological approach where species extinctions are mediated by the loss of the whole prey set of a consumer.” This contradicts the method description on row 156-157 saying that the authors follow the concept by Bellingeri & Bodini (2013) who considered that a species went secondarily extinct once is lost at least 25%, 50% or 75% of its original number of prey-interactions. The authors also states on row 197-198 that the extinction threshold represents the percentage of prey that a predator has to lose to go extinct. However, this is not the way the results are presented. I believe the authors define an extinction threshold as the number of interaction links left after a primary extinction (e.g. when a consumer has 0% of its interaction links left (i.e. no prey-interactions) (or 25%, 50% or 75%), than the consumer goes extinct). Please be consistent with your definition of extinction threshold and make sure that the text in the manuscript and figures describes the results following this definition. As now, this creates confusion.

3) On row 61-63, the authors briefly present a study (Marina et al 2018) that found the Potter Cove food web to be fragile to species loss, using topological analysis. Please explain the differences between this study and the current manuscript. For me, they show highly similarities.

Validity of the findings

Comments for improvements:
2) Add to the discussion of the differences between a topological approach and a dynamical approach simulating species loss. The authors mention, staring on row 307, that dynamics can influence the stability of food webs. Nevertheless, they do not discuss important mechanisms such as the lack of top-down cascades in topological methods and its influence on stability.
3) The conclusion of the study, presented on row 331, is to strong, saying that the results undoubtedly demonstrate the structural robustness of the Potter Cove to loss of macroalgae species. In face of the discussion part saying that the food web includes all nodes and interactions (without considering effects of seasonality and pulsatile nodes and interactions) in combination with the use of a topological approach known to provide more robust results compared to dynamical approaches, the conclusion should be presented in a more discrete manner.

Additional comments

I have now reviewed the manuscript by Cordone et al, entitled: Effects of macroalgae loss in an Antarctic marine food web: applying extinction threshold to food web studies. The authors present an analysis of food web robustness using a well-known method (topological extinction approach) which has been used to evaluate the impact on species loss in many food webs, making it difficult for this specific paper to fill a general knowledge gap. However, the robustness analysis has not, of course, been conducted for this specific food web of Potter Cove. The study is in its basic well done, but less well presented. As now, the result are not presented the way the method describes it causing confusion. I would not recommend this paper for publications as it stands today. However, I do believe the authors could make substantial improvements making the manuscript attractive to publication later on.

·

Basic reporting

I believe that the article is well structured and the subject of study is well explained.
The research question is showed to be relevant, well presented, and resolved.
The authors also do a good use of references to back their assessments and work.

As regarding the use of English, I would recommend a little revision of the English used in the document, since apart from some faults (ex. there is a missing "a" in the abstract as in "a hot biodiversity spot") the style sometimes is a little bit confusing.

The general structure of the article is well established and makes sense for the reader.
However I feel that some improvement should be done in the methods section to improve the clarity of how the experiment is carried upon. Also, I have some minor issues with the graphics that I think should be clarified to improve the readability of the manuscript.

In all, I believe it is a good research work. However, I feel I need to clarify how some of the work has been carried upon before accepting the manuscript.

Experimental design

My mayor issue is with the methods sections:
The most important is that even if the authors provide a good background of how are the topological extinctions performed in the cited literature, they do not provide a clear explanation of how they implement it. The points I think should be more clearly explained are:

-1) How do you obtain the “post-extinction” community ?:
In the manuscript you say : "Simulations consisted in remove(ing) target species from adjacency matrix deleting row and columns. Following every extinction simulation, a new adjacency matrix was computed, and several [quantities ...] were calculated", but How do you compute the new adjacency matrix? When do you consider that a species is extinct, when it has no more partners? Or when they are disconnected from the basal resources?
It is true that when doing extinctions in mutualistic networks it is enough to consider that a species undergoes extinctions once it has no more (or no more than a threshold) partners. However in trophic communities a species should go extinct not only if it has less than x partners, but also if it is no longer connected to the basal species in any ways (see Googling Food Webs: Can an Eigenvector Measure Species' Importance for Coextinctions? https://doi.org/10.1371/journal.pcbi.1000494).
I don't think your method of obtaining the post-extinction community it is clear enough in the manuscript as it is, and in order to make it more amenable to replication it should be further explained.

-2) How you perform the extinctions:
It is also confusing the way in which you carry on the extinctions. You say first “We defined different sequences of extinctions considering [topological properties ...]” , “After the loss of all macroalgae species, we removed the two sources of detritus” . So I understand from here that you erase ALL basal species in different order. If that is the case, then, how can it be possible that even if all basal species are lost there are still more than 100 links remaining in the community at the end of the extinctions? (see figure 3). Is it because there are still some basal species left? If that is, why do you attack the detritus but not these others? If, on the other hand you are attacking all basal producers, why do you still have a community remaining?

-3) Foodweb picture:
Finally I think that the manuscript could benefit from including the foodweb image in the main article and not it the supplementary. Since it is that community the one under attack it makes sense to me to show it in the article so readers can have a general idea of how it is. Maybe you could make a combined figure 2 with the common-enemy graph and show both at the same time?...

Validity of the findings

My first concern is with how the post-extinction community is obtained. I really think that in a foodweb one should consider that a species is extinct not only when it has no more prey, but also if it is no longer connected by any path to the basal producers. I'm not sure if the authors have taken this into account as I commented before ... In the case where they didn't I also believe that given the high connectance of the network the results should be similar even if this is taken into account. I suggest the authors should try to consider this in order to obtain a more realistic outcome of the topological extinctions.

My second concern regarding the results is with the last picture of figure 3, 4 and 5 (75% threshold image). In those images the x-range goes from 0 up to 60, when previously the authors have stated that they only have 40 species of macroalgae in the foodweb. How is it possible then to erase up to 60? If, on the other hand, what is reflected in the x-range is the percentage of attacked species it should be clarified in the figure someway. In that case, why do you erase only up to 40 or 60 %?

Finally I think that both the increasing and decreasing degree should be performed more than once due to the high degeneracy in the degree sequence (that is, many species have the same degree but they may have a different impact when erased from the community). I don't think it is going to have a real impact in the results at all but for the sake of completion I think it would be better to perform a similar study as they do with the random sequence, where they provide the average result and the deviations.

Additional comments

I've enjoyed reading the work, and I think it's really interesting.
However I believe these points should be clarified before publishing, to ensure maximum readability of the manuscript.

---

## Round 0.2 · Minor Revisions

Please consider the remaining minor revisions suggested by the reviewer.

·

Basic reporting

I've now read the new version of the manuscript, and I think that the authors have made a great improvement in the readibility of the manuscript and that it can be now accepted to be publisehd.

They have clarified the doubts I had with the methods and I believe now all readers can easily understand the procedure employed to perform the analysis.
The authors also made all the changes I suggested to the figures and I think they are more clear now, and all legends and axes are unambiguously described.
Also the authors present their results in perspective by mentioning how topological extinctions could overestimate the robustness of the foodweb, and taking into consideration the possible effect of temporal dynamics on the analysis.

All In all, I believe the authors have made a great effort in improving the quality of the manuscript, both by clarification of their methods and figures as well as the general writing, and I believe the article is ready to be published.

I'd like to note some minor faults I found during the lecture in case the authors would like to consider them:
line 33: Potter Cove food web is relative[ly] robust to macroalgae ...
line 47: is the most affected region of Antarctica due to climate change -> is the region most affected by climate Change
line 55: effects on [the] organisms of the cove
line 62: at [the] benthic
line 123: on [the] food web presented by Marina

Experimental design

The methods is clearly presented now in the supplementary information.

Validity of the findings

I believe there is a mistake in line 256:
The authors state : "When we compared the responses at different v, we found that an increase in v led to a decrease in the number of lost interactions (Lmin values of 211, 197, 129, and 30 for v of 0, 25, 50, and 75 %, respectively)."

I guess the authors mean that an increase in v leads to an INCREASE in the number of LOST interactions, as Lmin decreases when v is increased and vice-versa.

Additional comments

I'd like to thanks the authors for making the raw data and code available for all readers. I think sharing the methods and making the results easily reproducible is really important in science.

---

## Round 0.3 · accepted · Accept

Dear Georgina,

Your manuscript entitled "Effects of macroalgae loss in an Antarctic marine food web: applying extinction thresholds to food web studies" is accepted for publication in PeerJ.

#